# Binding to Iron Quercetin Complexes Increases the Antioxidant Capacity of the Major Birch Pollen Allergen Bet v 1 and Reduces Its Allergenicity

**DOI:** 10.3390/antiox12010042

**Published:** 2022-12-26

**Authors:** Andreas Regner, Nathalie Szepannek, Markus Wiederstein, Aila Fakhimahmadi, Luis F. Paciosis, Bart R. Blokhuis, Frank A. Redegeld, Gerlinde Hofstetter, Zdenek Dvorak, Erika Jensen-Jarolim, Karin Hufnagl, Franziska Roth-Walter

**Affiliations:** 1Comparative Medicine, The Interuniversity Messerli Research Institute of the University of Veterinary Medicine Vienna, Medical University Vienna and University of Vienna, 1210 Vienna, Austria; 2Department of Biosciences, University of Salzburg, 5020 Salzburg, Austria; 3Center of Pathophysiology, Infectiology and Immunology, Institute of Pathophysiology and Allergy Research, Medical University of Vienna, 1090 Vienna, Austria; 4Center for Plant Biotechnology and Genomics, Biotechnology Department, ETSIAAB, CBGP (UPM-INIA), Universidad Politécnica de Madrid, 28040 Madrid, Spain; 5Division of Pharmacology, Utrecht Institute for Pharmaceutical Sciences, Faculty of Science, Utrecht University, 3584 CG Utrecht, The Netherlands; 6Department of Cell Biology and Genetics, Faculty of Science, Palacky University, 78371 Olomouc, Czech Republic

**Keywords:** holo-Bet v 1, major allergen, immunomodulation, immune resilience, iron, quercetin, birch pollen, pathogenesis-related proteins, innate defense, nutritional immunity

## Abstract

Bet v 1 is the major allergen in birch pollen to which up to 95% of patients sensitized to birch respond. As a member of the pathogenesis-related PR 10 family, its natural function is implicated in plant defense, with a member of the PR10 family being reported to be upregulated under iron deficiency. As such, we assessed the function of Bet v 1 to sequester iron and its immunomodulatory properties on human immune cells. Binding of Bet v 1 to iron quercetin complexes FeQ2 was determined in docking calculations and by spectroscopy. Serum IgE-binding to Bet v 1 with (holoBet v1) and without ligands (apoBet v 1) were assessed by ELISA, blocking experiments and Western Blot. Crosslinking-capacity of apo/holoBet v 1 were assessed on human mast cells and Arylhydrocarbon receptor (AhR) activation with the human reporter cellline AZ-AHR. Human PBMCs were stimulated and assessed for labile iron and phenotypic changes by flow cytometry. Bet v 1 bound to FeQ2 strongly with calculated Kd values of 1 nm surpassing affinities to quercetin alone nearly by a factor of 1000. Binding to FeQ2 masked IgE epitopes and decreased IgE binding up to 80% and impaired degranulation of sensitized human mast cells. Bet v 1 facilitated the shuttling of quercetin, which activated the anti-inflammatory AhR pathway and increased the labile iron pool of human monocytic cells. The increase of labile iron was associated with an anti-inflammatory phenotype in CD14+monocytes and downregulation of HLADR. To summarize, we reveal for the first time that FeQ2 binding reduces the allergenicity of Bet v 1 due to ligand masking, but also actively contributes anti-inflammatory stimuli to human monocytes, thereby fostering tolerance. Nourishing immune cells with complex iron may thus represent a promising antigen-independent immunotherapeutic approach to improve efficacy in allergen immunotherapy.

## 1. Introduction

Not every protein is able to become under natural conditions an allergen. As such, allergens are clustered in protein families with the major allergens from mammalian systems always belonging to the “lipocalin” superfamily [1] and major allergens from plant origins usually belonging to either the pathogenesis-related proteins PR10 family or to the prolamin superfamily which comprises major allergens from the seed storage protein families being 2S albumins as well as the non-specific lipid transfer proteins nsLTPs [2,3]. Major allergens from plant origin also belong to the Gibberellin-regulated proteins GRPs [4], cupin protein superfamily (legumins-7S and vicilins-11S protein) and Ole e 1 families [5]. In 2012, our group made the serendipity discovery that the major birch pollen allergen Bet v 1 shares a lipocalin-like structure with the endogenous human lipocalin 2 LCN2 [6], which linked one of the largest major plant allergen family with a similar large major mammalian allergen family and with human LCN2. Indeed, LCN2 is decreased in allergic subjects [7] and as innate defense protein is usually present at mucosal surfaces and lymphoid sites. It is involved in numerous iron-dependent and immunoregulatory processes, where it can bind iron, but only when the iron is complexed by catechol-based siderophores, since lipocalins usually have no measurable affinity for iron alone [8]. LCN2′s ability to bind to iron makes it an antioxidant as it is able to prevent oxidative stress and prevent ferroptosis [9]. The iron-containing form of LCN2 (holoLCN2) is also present and released by anti-inflammatory macrophages [10] and contributes to tissue healing and recovery upon injury. In contrast, inflammatory macrophages release iron-free apo-LCN2 in response to invading bacteria [10].

In plants, immunity can be triggered via different pathways. One is the recognition via the recognition of pathogen-associated molecular patterns that leads to the induction of antimicrobial proteins, secondary metabolites, reactive oxygen species ROS formation and cell wall reinforcements. The second is the so-called effector-triggered immunity ETI, which can be initiated in plants by the perception of pathogen activity and usually leads to local program cell death named “hypersensitive response” (as a hallmark for systemic acquired resistance [11]). This ETI is triggered by salicylic acid accumulation to limit the pathogen spread [12]. Interestingly, iron depletion in plants is sufficient to prime plant immunity [13] leading to salicylic acid accumulation [13,14,15] and increasing coumarin and flavonoid synthesis [16,17,18,19] as well as the transcription of pathogenesis-related PR genes [15,20,21,22]. There seems to be a direct link between plant iron homeostasis and PR gene expression as iron treatment downregulates in plants the expression of PR10 proteins [23].

Interestingly, similar to mammalian lipocalins binding to iron via catechol-based siderophores, in 2014 the natural flavonoid ligand of Bet v 1 has been identified as quercetin-3-O-sophoroside [24], which likewise contains a catechol-moiety [25] with a very high affinity to iron [26]. As such, flavonoids seem to be the plant counterpart to bacterial-derived siderophores, with both considered secondary metabolites in the respected plants and bacteria/fungi and having both strong anti-oxidative and anti-inflammatory features, which are linked with their ability to complex iron.

In this respect, we hypothesized that similar to the mammalian lipocalin family, also in plants the natural function of the PR10 protein Bet v 1 is associated with plant iron homeostasis and immunity. We hypothesized that the allergenic or tolerogenic feature of Bet v 1 is linked to its iron-scavenging ability that it can exert cross-species, thereby enabling the modulation of the immune response in human immune cells with the iron-laden versus the iron-free form affecting the allergenicity of Bet v 1.

## 2. Materials and Methods

### 2.1. Structural and Docking Analysis

Atom coordinates of Bet v 1 were taken from the high-resolution (1.24 Å) crystal structure of its complex with naringenin (protein data bank (PDB) entry 4A87) [27]. The geometries of quercetin were obtained upon energy minimization with the MM2 force field of initial structures drawn using the ChemBioDraw/ChemBio3D Ultra 12.0 package. Docking input files for protein and ligands were prepared with reduce v3.23 [28], the ADFR software suite (https://ccsb.scripps.edu/adfr; accessed on 12 August 2021) and AutoDockTools [29]. Docking calculations were performed with AutoDock Vina [30,31]. The docking solution with the lowest affinity energy E_aff_ was selected. Estimates of dissociation equilibrium constants K_d_ were then calculated for the protein-ligand complexes by assuming E_aff_~ΔG with K_d_ = exp(−ΔG/RT) at T = 298.15. Protein structural visualizations were prepared with UCSF Chimera [32,33].

### 2.2. Recombinant Bet v 1

Bet v 1 was produced as described [34]. In short, a codon-optimized synthetic gene of Bet v 1.0101 was obtained from Eurofins MWG Operon (Ebersberg, Germany) and cloned into the expression vector pET-28a(+) (Merck Millipore, Darmstadt, Germany). The proteins were expressed in *E. coli* BL21[DE3] in LB medium at 37 °C after induction with 1 mM isopropyl--D-thiogalactopyranoside. Bet v 1 was purified by a combination of hydrophobic interaction and ion exchange chromatography. SDS-PAGE, MALDI-TOF MS (Bruker Ultraflex II, Bruker Daltonics, Billerica, MA, USA), and circular dichroism (CD) spectroscopy was used to verify protein purity and identity, mass, and secondary structure. Measurement of endotoxin content was done by Hyglos EndozymeII Kit (Bernried am Starnberger See, GER) and total protein content by BCA assay according to the manufacturer’s instructions (Pierce BCA Protein Assay Kit, Thermo Scientific, Rockford, IL, USA).

### 2.3. Generation of Apo- and Holo Bet v 1

Bet v 1 (1.1–1.89 mg/mL) was dialyzed three times against 10 µM deferoxamine mesylate salt (Sigma D9533, St. Louis, MO, USA) following dialyzation against deionized water to generate apoBet v 1. HoloBet v 1 was generated by adding pre-formed iron-quercetin complexes (FeQ2). 3 mM FeQ2 complexes were generated by dissolving quercetin (Sigma 1592409) in 1 M NaOH and adding acidic iron (Iron-standard AAS, Sigma 16596) at a ratio 2:1, before adding them to apoBet v 1 to a final concentration of, e.g., 10 µM Bet v 1, 20 µM quercetin and 10 µM iron or dilutions thereof.

### 2.4. Spectral Analysis

For spectral analysis, a physiological saline solution (0.89% NaCl) was used as a buffer to minimize iron contamination from the air. The pH was kept constant at pH 7 for Figure 1c,d and was >7.3 for Figure 1e–g. Optical density was measured using (1) 200 µM quercetin (=100 µM Q2) with increasing concentrations of ferric iron (40–200 µM), (2) 100 µM FeQ2 with increasing concentrations of Bet v 1 (2–20 µM) or (3) 100 µM FeQ2 with increasing concentrations of gel-filtrated apoBet v 1 or holoBet v 1 (0.5–8 µM). All measurements were repeated at least three times with similar results.

### 2.5. Anti-Oxidative Assay

Anti-oxidative status was measured as described before [35,36]. Shortly, a 7 mM ABTS radical stock solution was prepared by diluting 19 mg of 2,2′-Azinobis-(3-ethylbenzothiazoline-6-sulfonic acid) (Sigma A1888) (ABTS+.) and 20 mg ammonium persulfate (Sigma A3678) in 5 mL distilled water at room temperature. The stock solution was further diluted 1:4 in distilled water to obtain an ABTS working solution. As standard either Trolox (Sigma 238813, Austria, St. Louis, MO, USA), quercetin Q2 or FeQ2 was used. Samples of apoBet v 1 (0.5 mg/mL) and/or preincubated with FeQ2, Q2 or ferroxamine (FO) were subjected to gel filtration (PD MiniTrap G-25 Columns, GE Healthcare 28-9180-07) to remove unbound ligands. 50 µL of diluted samples or diluted standards (Trolox, Q2, FeQ2) were incubated with 50 µL of ABTS working solution for 2 min in a 96 well plate before measurement of absorbance at 740 nm using an Infinite M200Pro microplate reader (Tecan, Grödig, Austria). Data are shown as antioxidative capacity using Trolox-equivalent normalized to apoBet v 1 in Figure 1f,g using FeQ2 and Q2 as standard to calculate the molar ratio of quercetin to Bet v 1.

### 2.6. Bet v1-Specific Sera

Serum samples derived from two clinical trials (NCT0381598, NCT03816800), which were approved by the ethics committee (#1972/2017 and #1370/2018) of the Medical University of Vienna. Patients’ characteristics have been described previously in detail [37,38]. Briefly, sera of subjects with allergic rhinitis, but were otherwise healthy, were used with Bet v 1-specificity of patients to ascertain by testing for specific IgE by ALEX^®^-tests (Macroarray Diagnostics, Wien, Austria) and ALEX^®^-test specific IgE-level against Bet v1 being at least 1 kU_A_/L. The concentration of individual sera and in the pools for Bet v 1-specific IgE ranged from 10 to 60 kU_A_/L.

### 2.7. Bet v1-Specific IgE ELISA

Two µg/mL of apoBet v 1, holoBet v1-FeQ2, Bet v 1-Q2 or Bet v 1-FO (ferroxamine) diluted in 0.89% NaCl were coated (100 µL per well) overnight at 4 °C, blocked for 2 h at room temperature with 1% BSA in 0.89% NaCl containing 0.05% Tween 20 (200 µL per well), before incubated with 100 µL of diluted serum from birch-pollen allergic and non-allergic subjects (1:10 diluted in 0.89% NaCl/0.05% Tween-20) overnight at 4 °C. Bound IgE was detected with horseradish–peroxidase-conjugated goat anti-human IgE antibody (Invitrogen A18793, Waltham, MA, USA) diluted at 1:4000 in 0.89% NaCl containing 0.05% Tween-20 and using tetramethylbenzidine (eBioscience, San Diego, CA, USA) as a substrate. Color development was stopped with 1.8 M sulfuric acid. The optical density was measured at 405 nm by using an Infinite M200Pro microplate reader (Tecan, Grödig, Austria). Between the steps, rigorous washing was performed with 0.89% NaCl containing 0.05% Tween-20. Data are shown as a percentage normalized to apoBet v 1 IgE binding levels or as OD at 450 nm.

### 2.8. Bet v 1-Specific IgE Inhibition ELISA

Inhibition experiments were carried out similarly as described for the Bet v 1-specific IgE ELISA. Briefly, plates were coated with 2 µg/mL of apoBet v 1 or holoBet v 1-FeQ2 (100 µL per well) overnight at 4 °C. In the meantime, pooled serum from birch-pollen allergic subjects (diluted 1:10 in 0.89% NaCl) was preincubated with increasing doses of apoBet v 1 (0–1000 ng/mL) or holoBet v 1-FeQ2 (0–1000 ng/mL) also overnight at 4 °C. After washing and blocking, preincubated serum samples were applied (100 µL per well) overnight at 4 °C. Detection was performed by using horseradish–peroxidase-conjugated goat anti-human IgE antibody (Invitrogen A18793) diluted at 1:4000 in 0.89% NaCl containing 0.05% Tween-20, with use of tetramethylbenzidine (eBioscience) as a substrate and 1.8 M sulfuric acid to stop color development. The optical density was measured at 450 nm with the Infinite M200Pro microplate reader (Tecan, Grödig, Austria). Data are shown as OD at 450 nm or as kU/L IgE binding to plate-bound apoBet v 1.

### 2.9. SDS-PAGE and Western Blot

ApoBet v 1 and holoBet v 1-FeQ2 were mixed with 4x non-reducing sample buffer and 12 µg/slot was applied on two sodium dodecyl sulfate polyacrylamide (SDS–PAGE) 4–20% gradient gels (Biorad 4568095, Tokyo, Japan), run at 90 V for about 1 h and subsequently one gel was stained with Roti-Blue quick solution (Roth 4829.2, Roth, Germany) and the other gel underwent immunoblotting. Briefly, the gel was blotted on a methanol-activated PVDF-membrane (Immobilon IPVH00010, 0.45 µm), blocked with 1% BSA in tris buffered saline containing 0.05% Tween-20 (TBS-T) for 18 h, before incubating with 1:10 diluted serum pool of allergic donors (*n* = 17) for 18 h at 4 °C. Bound IgE was detected with horseradish peroxidase labeled anti-human IgE antibody (Invitrogen A18793) and using ECL substrate (clarity TM Western ECL Substrate, BioRad 170-5061) for detection. Between each step, rigorous washing was performed with TBS-T. Luminescence was imaged with the ChemiDoc™Touch Imaging System (BioRad).

### 2.10. Human Mast Cell Degranulation Assay

Human peripheral blood mononuclear cell-derived mast cells were generated as previously described by Folkerts et al. [39]. Briefly, peripheral blood mononuclear cells were obtained from buffy coats of healthy blood donors and CD34+ precursor cells were isolated using the EasySep Human CD34 Positive Selection Kit (STEMCELL Technologies). CD34+ cells were maintained for 4 weeks under serum-free conditions using StemSpan medium (STEMCELL Technologies) supplemented with recombinant human IL-6 (50 ng/mL; Peprotech), human IL-3 (10 ng/mL; Peprotech) and human Stem Cell Factor (100 ng/mL Peprotech, Rocky Hill, NJ, USA). After 4 weeks, the cells were cultured in Iscove’s modified Dulbecco’s medium/0.5% bovine serum albumin with human IL-6 (50 ng/mL, Peprotech, Rocky Hill, NJ, USA) and 3% supernatant of Chinese hamster ovary transfectants secreting murine stem cell factor (a gift from Dr. P. Dubreuil, Marseille, France). The mature MCs were identified by flow cytometry based on positive staining for CD117 (eBioscience) and FcεRIa (eBioscience) using BD FACS Canto II (approximately 90%). Degranulation assay was performed by incubation of human mast cells with serum pools of birch pollen allergic or non-allergic subjects followed by incubation with 5 nM of apoBet v 1, holoBet v1, Q2, FeQ2 or buffer, respectively. Degranulation was assessed by measurement of the activity of released ß-hexominidase in the supernatant and unreleased enzyme in the respective cell lysate. The presented results were calculated as percentage released versus total ß-hexominidase activity, with a release from unstimulated controls being 0.003%, from positive controls with anti-human IgE being 33.4% and with ionomycin being 84.6%.

### 2.11. AZ-AhR Reporter Assay

AZ-AhR assay was done as previously described [40]. Briefly, AZ-AhR cells were plated on 96-well plates at a density of 2 × 10^4^ cells/well for 18 h. Subsequently, cells were stimulated for 18 h in triplicates with 45 µM of iron-quercetin complexes alone or in addition with 2 µM or 10 µM Bet v 1. The positive control cells were treated with 20 nM indirubin. Cells were washed once with 0.89% NaCl, before the lysis buffer of the luciferase assay kit (Promega E4530, Tokyo, Japan) was added. After a single freeze–thaw cycle, 20 μL/well of lysates were transferred into a black 96-well flat-bottom plate (Thermo Scientific) and bioluminescent reaction were started with addition of 100 μL/well of luciferase assay reagent (Promega). Chemiluminescence was measured (10 s/well) using a spectrophotometer (Tecan InfiniteM200 PRO). Data from six independent experiments are shown as normalized to medium levels.

### 2.12. Flow Cytometric Analysis of Human PBMCs

PBMCs were isolated by Ficoll-Paque (GE Healthcare) and washed with 0.9% NaCl before incubation with 5 µM iron-quercetin (FeQ2) alone and/or in combination with 5 µM apoBet v1 in media containing neither phenol red nor FCS for 18 h as previously described [41]. Subsequently, cells were stained with combinations of calcein-AM (Thermo-Fisher, Waltham, MA, USA), CD3-APC-Cy7 (eBioscience, clone SK7), CD14-APC-Cy7 (Biolegend, clone M5EZ), HLADR-PE (Biolegend, San Diego, Calif, clone L243PC), and CD86-PE-CY7 (Biolegend, clone IT2.2) for flow cytometric analysis. Calcein-AM violet was used as a living marker and to determine the labile iron load in living cells. Doublets were excluded before gating on CD3+ cells in the lymphocyte population or on CD14+ cells in the monocytic population. In monocytic cells this was followed by consecutive gating for HLADR+, CD86+ and calcein+ and geometric mean fluorescence intensity (MFI) was calculated for each fluorochrome. Acquisition and analyses were performed on a FACS Canto II machine (BD Bioscience, San Jose, CA, USA) using the FACSDiva Software 6.0 (BD Biosciences).

### 2.13. Statistical Analyses

Anti-oxidative assay and mast cell degranulation were compared with one-way ANOVA following Holm-Sidák’s multiple comparisons test, with a single pooled variance. Data from IgE ELISA were compared with RM one-way ANOVA with Geissner Greenhouse correction and Holm-Sidák’s multiple comparisons test and inhibition ELISA with two-way ANOVA using Holm-Sidák’s multiple comparisons test, with a single pooled variance. AhR activation and flow cytometry of PBMCs was analyzed by mixed-effect analysis or with RM one-way ANOVA including Geisser-Greenhouse correction followed by Tukey’s multiple comparisons test. All tests were considered significant when *p* < 0.05. Statistical analysis was performed with Graphpad prism 9.4.1 (San Diego, CA 92108, USA).

## 3. Results

### 3.1. Bet v 1 Binds Strongly to Iron-Quercetin Complexes In Silico and In Vitro

The natural ligand of Bet v 1 has been isolated with a quercetin-core structure [24], with quercetin known to act as a very strong iron chelator [42]. The binding of this ligand transforms Bet v 1 from apo- (empty) to holo-Bet v 1. Quercetin alone has anti-inflammatory properties itself, and from previous studies [6,41,43], we hypothesized that the binding of this immunomodulatory molecule in conjunction with iron would modify the allergenicity of Bet v 1. In the first step, we proved by different means that Bet v 1 binds to iron-quercetin complexes (FeQ2). In silico analysis revealed that Bet v 1 binds by a factor of 10^3^ stronger to FeQ2 compared to quercetin alone by (Figure 1a,b). The calculated FeQ2 binding affinity energy was −12.092 kcal/mol, corresponding to a dissociation constant Kd of 0.0013 µmol/L, whereas binding to quercetin alone has an affinity energy of −8.139 kcal/moL, corresponding to a Kd of 1.0709 µmol/L.

Additionally, by spectroscopical means iron-quercetin complex formation was visible (Figure 1c). Binding to FeQ2 could be proven, because a second absorption maximum diminished concentration-dependently upon addition of apoBet v 1 (empty protein, no ligand) (Figure 1d), but not when holoBet v 1 (protein-ligand complex) was added (Figure 1e). After removing potential unbound ligands by gel filtration, we show that the antioxidative capacity of Bet v 1 increased due to binding to the ligands FeQ2 and quercetin, which allowed also to compute the number of bound ligands per Bet v 1 molecule, being 1.29 ± 0.39 for FeQ2 complexes and 0.90 ± 0.12 for quercetin (Figure 1f,g). To sum up, we provide evidence, that Bet v1 strongly binds to iron-quercetin complexes at physiological relevant pH.

### 3.2. IgE Epitopes Masking and Less Mast Cell Degranulation by Holobet v 1

In the next step, we investigated whether the allergenicity of Bet v1 was affected by binding to iron—quercetin complexes. Bet v1 has two described IgE epitopes to which birch pollen allergic IgE antibodies bind to spanning, from aa29 to aa58 (epitope 1) and from aa73 to aa103 (epitope 2) [44]. In silico analysis revealed that both IgE epitopes were affected by FeQ2: epitope 1 at Phe 30 and Phe 58 and epitope 2 at Tyr 81, Tyr83 and Ile 102. (Figure 2a) similar as described for FeQ2 binding masking major lipocalin allergens [40,41]. Next, we verified the biological relevance of the in silico findings, by testing whether individual IgE of birch pollen allergic patients (*n* = 12) bound similarly to plate-bound apoBet v1 (without any ligands) or with the ligands FeQ2 (Betv1_FeQ2), quercetin alone (Betv1_Q2) or using ferroxamin complexes as a hydroxame-based siderophore-iron complex to which Bet v1 should not bind to (Betv1_FO) as control. Indeed, a significant reduction in IgE-binding was very apparent against holoBet v 1 with FeQ2 and to a lesser degree also to quercetin alone (Figure 2b). Additionally, in blocking experiments with pooled sera preincubated with various doses of apoBet v 1, the binding to plate-bound holoBet v 1 was significantly reduced compared to apoBet v 1 (Figure 2c). In blocking experiments when individual sera (*n* = 6) were preincubated with various concentrations of apo- or holo-Bet v 1 a significantly greater blocking (Appendix A). Thus reduced IgE-binding was achieved with the apo-form—as depicted in the summary graph and confidence interval plot in Figure 2d—and highlights that IgE epitopes were masked and not accessible with the holo-form (Figure 2d). A similar pattern was evident in Western Blot, with a reduced IgE-binding (using pooled sera) seen when Bet v 1 was in conjunction with FeQ2 (Figure 2e). Moreover, when human-generated mast cells were sensitized with sera of birch pollen allergic sera, crosslinking with holoBet v1 compared to apoBet v 1 was less efficient and resulted in significantly less degranulation. No degranulation was achieved when mast cells were sensitized with non-allergic sera, (Figure 2f,g).

### 3.3. Bet v 1 Facilitates Quercetin-Dependent AhR Activation—And Iron Transport into Human Cells

Next, we aimed to prove that the ligands were responsible for the reduced allergenic features of Bet v 1 and that in fact the uptake of these ligands into human immune cells was facilitated by Bet v 1. To prove that Bet v 1 facilitated quercetin-transport into human cells, the human AZ-AhR cell line stably transfected with several aryl hydrocarbon-receptor AhR binding sites upstream of the luciferase reporter gene [45] were stimulated with apo-/holo-Bet v 1, as quercetin is a known activator of AhR. Indeed, the activity of AhR concentration-dependently increased by the addition of Bet v 1 from 3.0 ± 1.4 with FeQ2 alone to 5.6 ± 2.6 with 2 µM and to 10.84 ± 7.1 with 10 µM Bet v 1 (Figure 3a). To proof that not only quercetin, but also iron was transported into human immune cells, PBMCs from allergic donors (*n* = 34) were stimulated with apo/holoBet v 1. Indeed, though Bet v 1 led to an increase in cell survival (Appendix A) in the lymphocytic compartment and led to the relative expression of CD3+, this was irrespective of the ligand and also the labile iron pool, measured by the quenching of the Calcein-signal—was not affected (Figure 3b,c). In contrast, relative CD14 expression decreased upon Bet v 1 exposure but further significantly decreased in PBMCs incubated with holoBet v 1 compared to cells stimulated with apoBet v 1 only. Moreover, holoBet v 1 incubation led to a significant increase of the labile iron in monocytes with PBMCs being unaffected by apoBet v 1 and also by incubation of FeQ2 alone (Figure 3d,e). Importantly, the relative numbers of living cells did not differ between apo- and holoBet v 1 incubated cells (Appendix A). To sum up, here we show that Bet v 1 was able to shuttle quercetin as well as iron into human cells, thereby on the one hand activating the anti-inflammatory AhR pathway and on the other hand increasing the labile iron of CD14+ cells, but not of CD3+ cells.

### 3.4. Bet v 1 Acts Immunomodulatory by Importing Iron and Suppressing Antigen Presentation

We further assessed the impact of iron-shuttling by Bet v 1 on antigen presentation by assessing labile iron and the expression of HLADR as well as the costimulatory molecule CD86+ in CD14+ cells. Along with the increase of labile iron also the maturation state of the CD14+ cells changed with a significantly less mature state of CD14+HLADR+CD86+ cells observed in holoBet v 1-stimulated PBMCs (Figure 4a,b). Particularly CD14+ with HLADR^low/−^ are attributed to an immunosuppressive phenotype [46]. An increase of the labile iron was especially associated with a decrease in HLADR expression, which was not seen with CD86 (Figure 4c,d). As such, relative numbers of mature monocytes with antigen presenting skills were associated in regression analyses with a decrease of labile iron (Figure 4e).

Hence, we show that shuttling of labile iron by Bet v 1 strongly affected the maturation state of monocytic cells keeping them immature and repressing antigen presentation.

## 4. Discussion

The present study supports our notion that the natural function of Bet v 1 is strongly linked with its ability to bind to complexes of iron, a redox-active element essential for most life [43]. The strong inherent affinity of particularly ferric iron to flavonoids/polyphenols is highlighted by the fact that flavonoid-iron complexes can self-assembly into huge polyphenol-metal networks [47,48], which is used in the industry to coat microorganisms, but also by the fact that solubilized iron-flavonoid complexes serve as an iron source for commensal and opportunistic microbial pathogens [42]. Importantly, in plants intracellular iron depletion is sufficient to lead to the expression of pathogenesis-related proteins [15,20,21,22], further emphasizing that the natural function of Bet v 1 is linked with plant defense and nutritional immunity (particularly of iron), similarly as the mammalian counterpart LCN2. Indeed, using the cow milk lipocalin protein beta-lactoglobulin, we could show in previous in vivo [40,49] and in clinical studies [38,50,51] that nutritional support to immune cells was essential to induce immune resilience, amelioration of symptoms and tolerance in allergic rhinitis patients.

Here, we show similarly, that the binding affinity to quercetin-iron by Bet v 1 is exceptionally high with calculated affinity being in the lower nM-range, which is about 1000× stronger than to quercetin alone. Moreover, we show that binding to iron-quercetin complexes was superior to quercetin alone in hampering epitope-recognition by specific IgE antibodies from birch pollen-allergic individuals. Importantly, this had also implications for the effector phase as crosslinking by holoBet v 1 was not as effective as with apoBet v 1 without any ligands, in which epitope masking did not hinder crosslinking. It also indicates that during allergic sensitization Bet v 1 was presented to the human immune system without any ligand, as otherwise the generation of IgE against these epitopes would have been impeded.

In our study, the impact of the iron-quercetin ligand on rendering Bet v 1 less allergenic was not restricted on IgE recognition. Indeed, here we give evidence that the carrier function of Bet v 1 is crucial to provide an anti-inflammatory and anti-oxidative stimulus to human immune cells. Both the transport of quercetin as well as iron was facilitated by Bet v 1 with quercetin able to activate the arylhydrocarbon receptor, but also to increase the labile iron pool in human monocytic cells, in which a large labile iron pool is a major characteristic of anti-inflammatory macrophages [43].

Activation of the cytoplasmic promiscuous AhR, which interacts with a plethora of exogenous ligands including quercetin [52], is described to mediate anti-inflammatory stimuli [53] that promote regulatory T [54,55], but not Th2 [56] cell differentiation and, keep antigen-presenting cells such as dendritic cells or macrophages in an immature state [57].

Iron seems to have a similar impact- when not present in a free form to generate reactive oxygen species- with an increase of the labile iron content in macrophages promoting an anti-inflammatory phenotype and iron depletion promoting an inflammatory phenotype [58,59]. Indeed, iron deficiency is associated in vitro [60,61,62] and in human clinical studies with Th2 inflammation [63,64,65,66,67,68,69]. Iron is also central to the human immune defense, which is highlighted by the fact that the macrophages are the central hub for iron handling, recycling and distribution [70]. Importantly, iron distribution and recycling stops when the macrophage changes due to immune activation to an inflammatory phenotype. As such, both too much and too low intracellular iron levels have been reported to cause oxidative damage in mitochondria and trigger the expression of inflammatory-related genes [71]. This also explains that in humans every chronic immune activation will lead over time to anemia of chronic inflammation [43]. Despite the phylogenetic distance between plants and human, the innate immune mechanisms are evoked by the same triggers, with not only humans being affected by a hypersensitive response towards allergens, but also plant mounting a hypersensitive response that leads to the expression of pathogenesis-related proteins in response to iron sequestration. Although this immune priming is a very desirable response to infections, it also turns otherwise harmless proteins into allergens.

Our current understanding indicates that the major birch pollen allergen Bet v 1—as a prime example of pathogenesis-related proteins [72]—has the ability to sequester iron, similarly to the mammalian lipocalin family is able to bind iron, and that exactly this feature makes them potent allergens. When they do not carry ligands, they are able to deprive their surrounding of iron complexes, thereby actively leading to immune activation and inflammation [70]. In contrast, when they do carry iron, they indeed are able to augment the labile iron pool in macrophages and thereby promote an immature, tolerogenic and anti-inflammatory phenotype in macrophages [58].

## 5. Conclusions

To summarize, we reveal for the first time that FeQ2 binding reduces the allergenicity of Bet v 1. We provide evidence that both iron and quercetin are shuttled by Bet v 1 into human immune cells thereby hindering the activation of monocytes by increasing the labile iron levels. The present data are in line with our findings that Bet v 1 can transport micronutrients and thereby promotes tolerance. As such, the principle of nourishing immune cells with micronutrients in an antigen-non-specific manner may be one immunotherapeutic approach to improve efficacy.

## 6. Patents

FRW, LFP and EJJ declare inventorship of EP2894478 (Roth-Walter F et al., Method and means for diagnosing and treating allergy.) (applicant Biomedical International R+D GmbH, Vienna, Austria). EJJ declares shareholdership in Biomedical Int. R+D GmbH, Vienna, Austria.

## Figures and Tables

**Figure 1 antioxidants-12-00042-f001:**
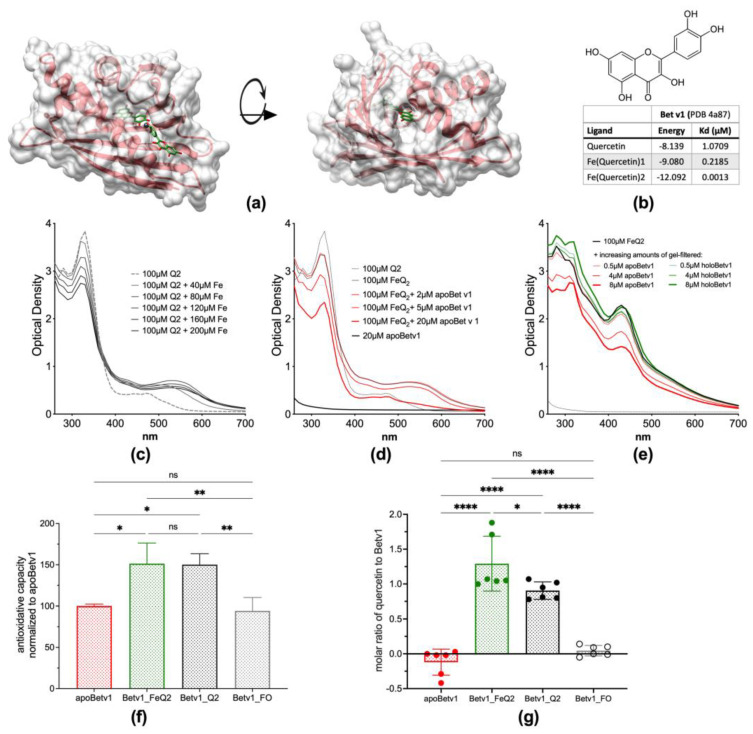
Bet v 1 binds iron—quercetin complexes (**a**): Protein surface of the major birch pollen allergen Bet v 1, incorporating iron(quercetin)_2_ (FeQ_2_) (sticks with carbons in green, oxygens in red, and iron shown as a grey sphere) (**b**): Molecular structure of quercetin and calculated affinities of quercetin in conjunction with iron to Bet v 1. (**c**): Optical spectra of 200 µM quercetin (=100 µM Q2) with increasing concentration of ferric iron at pH 7.3. (**d**): Optical spectra of 20µM Bet v1, 100 µM Q2 or FeQ2 alone, or by adding increasing concentrations of Bet v 1. (**e**): FeQ2 spectra with and without the addition of increasing doses of gel—filtrated purified apoBet v 1 or holoBet v 1. (**f**): ApoBet v 1 was preincubated with FeQ2, Q2 or ferroxamine (FO), before removing unbound ligands by gelfiltration and performing a Trolox Equivalent Antioxidative test using Trolox as reference. (**g**): Calculated amount of bound quercetin to Bet v 1 measuring the antioxidative capacity with quercetin and FeQ2 as reference. For statistical analyses (**f**,**g**) were compared with one-way ANOVA following Holm-Sidák’s multiple comparisons test with a single pooled variance; Mean ± SD; ns not significant, * *p* < 0.05, ** *p* < 0.01, **** *p* < 0.0001.

**Figure 2 antioxidants-12-00042-f002:**
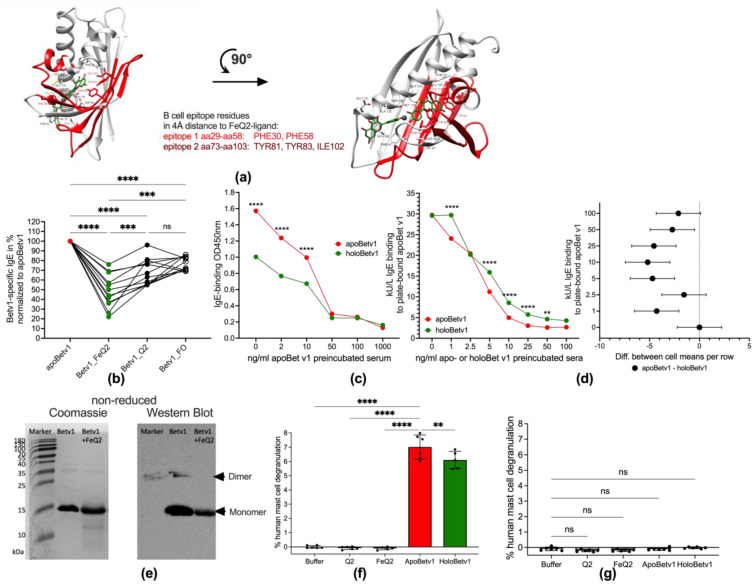
Binding of iron—quercetin complex by Bet v 1 masks B cell epitopes and affects IgE binding and mast cell degranulation (**a**): Structure of Bet v 1—Fe(quercetin)_2_ complex shown as cartoon from two different views. Iron complexed by two quercetins (FeQ2) is depicted as sticks with carbons in green and oxygens in red. Iron atom (Fe) is represented as a grey sphere. Major B-cell epitope 1 (29–58) and epitope 2 (73–103) are marked in light and dark red, respectively. Residues within a 4Å distance from any atom of FeQ2 are shown as sticks with carbons in red. (**b**): Serum IgE binding of birch pollen allergic (*n* = 12) to Bet v 1 without cargo (apoBet v 1) or in combination with iron-quercetin (Bet v 1_FeQ2), quercetin (Bet v 1_Q2), or ferroxamine (Bet v 1_FO). (**c**): Blocking experiments with pooled birch pollen allergic incubated with increasing doses of apo Bet v 1 and binding to plate-bound apo- (red line) or holoBet v 1 (green line). (**d**): Summary and confidence interval plot of blocking experiments incubating 6 individual sera of birch pollen allergic subjects with increasing doses of apo- or holoBet v 1 and assessing IgE binding to plate-bound apoBet v 1. (**e**): Protein bands stained by Coomassie and serum IgE binding against Bet v 1 alone or in combination with FeQ2 of gels run under non-reducing conditions. Degranulation of human mast cells sensitized with pooled (**f**) birch pollen allergic (**g**): non-allergic sera. For statistical analyses (**b**) was compared with RM one-way ANOVA with Geissner Greenhouse correction and Holm-Sidák’s multiple comparisons test (**c**,**d**) with two-way ANOVA using Holm-Sidák’s multiple comparisons test, with a single pooled variance, (**f**) with mixed-effect analyses using the Geissner Greenhouse correction and Holm-Sidák’s multiple comparisons test, (**g**) with one-way ANOVA following Holm-Sidák’s multiple comparisons tes. Mean ± SD; ns not significant, ** *p* < 0.01, *** *p* < 0.001, **** *p* < 0.0001.

**Figure 3 antioxidants-12-00042-f003:**
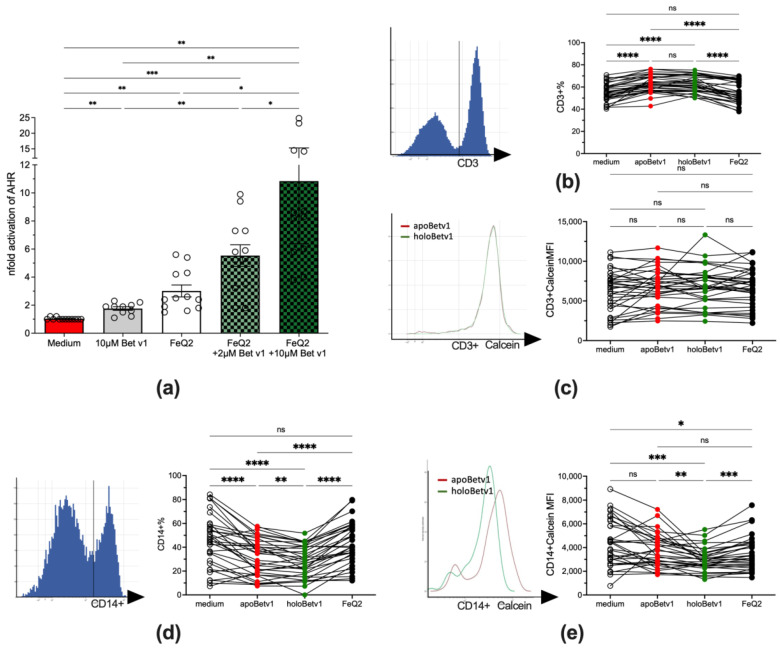
Shuttling of FeQ2 by Bet v 1 activates the anti-inflammatory aryl hydrocarbon receptor and increases labile iron in monocytes. (**a**): AhR activation by quercetin-iron complexes is increased upon addition of Bet v 1. AZ-AhR cells were treated with 45 µM of iron-quercetin complexes alone or with 2 µM and 10 µM Bet v 1, for 18 h before luciferase-activity was measured in the supernatant. Summary from six independent experiments normalized to medium alone. Flowcytometric analyses of PBMCs (*n* = 34) stimulated with medium, iron-quercetin (FeQ2), apoBet v 1 alone or in combination thereof (holoBet v 1). (**b**): Gated CD3+ cells and summary of relative CD3+ expression and (**c**): mean fluorescence intensity (MFI) of the calcein signals thereof. (**d**): Gating and summary of relative numbers of CD14+ expression as well as (**e**) MFI of the calcein signals of stimulated CD14+ cells. An increase of intracellular iron results in a lower calcein-signal due to quenching. Data were analyzed in (**a**,**d**,**e**) by mixed-effect analysis and (**b**,**c**) with RM one-way ANOVA. All data were Geisser-Greenhouse corrected and followed by Tukey’s multiple comparisons test. * *p* < 0.05; ** *p* < 0.01; *** *p* < 0.001; **** *p* < 0.0001; ns = not significant.

**Figure 4 antioxidants-12-00042-f004:**
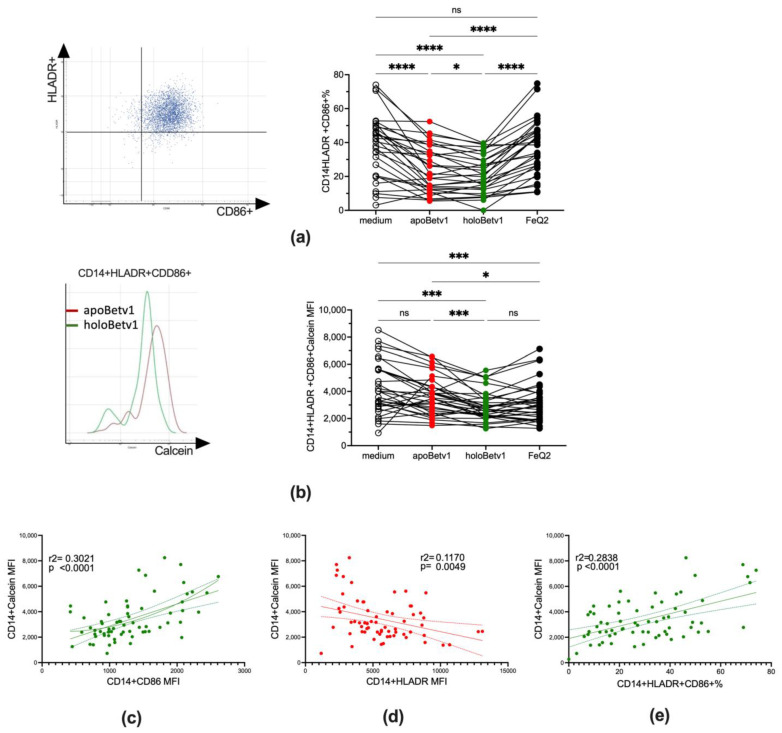
HoloBet v 1 decreased antigen presentation is associated with an increase of labile iron in monocytic cells. Flowcytometric analyses of PBMCs (*n* = 34) stimulated with medium, iron-quercetin (FeQ2), apoBet v 1 alone or in combination thereof (holoBet v 1). (**a**): gating strategy and relative numbers of CD14+HLADR+CD86+ in the monocytic gate and (**b**): the respected MFI of the calcein signal. Linear regression analysis of the Calcein signal of CD14+ cells (**c**): versus CD86 expression and (**d**): HLADR expression as well as (**e**): calcein-signal of CD14+ with the relative numbers of CD14+HLADR+CD86+ cells. Data were analysed in (**a**,**b**) by mixed-effect analysis. All data were Geisser-Greenhouse corrected and followed by Tukey’s multiple comparisons test. (**c**–**e**). Simple linear regression models of data with 95% confidence bands are depicted. * *p* < 0.05; *** *p* < 0.001; **** *p* < 0.0001; ns = not significant.

## Data Availability

The data presented in this study are all contained within the article.

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
