# Peer review of "Binding to Iron Quercetin Complexes Increases the Antioxidant Capacity of the Major Birch Pollen Allergen Bet v 1 and Reduces Its Allergenicity"

_antioxidants, 2022, doi:10.3390/antiox12010042_

Round 1
Reviewer 1 Report (Previous Reviewer 2)
The manuscript was previously revised by 3 reviewers. Two accepted and one rejected publication. The authors corrected and answered the reviewers comment and I decide accept the manuscript for publication.
Reviewer 2 Report (Previous Reviewer 1)
Corrections done

This manuscript is a resubmission of an earlier submission. The following is a list of the peer review reports and author responses from that submission.
Round 1
Reviewer 1 Report
The paper clearly demonstrates that Fe-Q2 is amore avid ligand for Bet v 1. This is of interest for both the possible reduced allergenicity, the anti-inflammatory effects and the theory of Fe augmented sensitisation.
The extent to which the binding reduces IgE binding not clear. There is a range evenly distributed from 20-80% in the normalised data in Figure 2b. It is not known if there is baseline effect of the different binding capacity of the different forms of Bet v 1 to the ELISA plates. The wobbly curve in figure 2d would suggest a 2-5 fold decrease but this is done with a pool and the effect size might be very largely due to a single high-titre sera
The titre of anti-Bet v IgE might be critical and is not shown. Overall it would be more informative and convincing for me to present a series of inhibition curves as per Figure 2d but for individual sera and with the y-axis showing IU or ng/ml of IgE binding. There is no shortage of Bet v 1 allergic subjects in Austria (as regularly expounded in the authors papers) and it does not look as if there is a shortage of reagents or workers. Even about six individual graphs would be good.
The mast cell degranulation does examine individuals. The difference between the individuals is nowhere as varied as the the IgE binding and there is not much of a difference between the different forms of Bet v 1. This should be discussed.
In light of the results the "and renders it hypoallergenic" in the title would best be changed to "and reduces allergenicity". The abstract similarly needs change,
Somewhere in the paper the the IgE titres of the patients should be given and the type of disease.
Reviewer 2 Report
The authors found out earlier that specially treated (deferoxamine mesylate) Bet v 1 protein can form complexes with iron. They used this property to form complexes with antioxidant quercetin. They demonstrated some antioxidant activity, more in aryl hydrocarbon receptor gene expression. The main result was the demonstration of a decrease in IgE binding.
Title and abstract
Both the title and aim of this MS are slightly unfocused. The title: “Binding to iron quercetin complexes increases the antioxidant capacity of the major birch pollen allergen Bet v 1 and renders it hypoallergenic”.
The title says that:
1) iron quercetin complexes do something….however total complex Betv1-deferoxamine-mesylate-iron-quercetin does something;
2) “…. increases the antioxidant capacity of the major birch pollen allergen Bet v 1…”; the antioxidant activity not of Bet v 1 but of quercetin; moreover neither Betv1 nor iron quercetin separately were not included as controls (Fig.1);
3) “…renders it hypoallergenic ….” This can be tested only in vivo as there is a competition for iron in body. Antioxidant capacity is not important for ASIT. It looks like the authors combined two activities with the opposite properties.
Possible title: Antioxidant activity and IgE binding of Bet v 1-iron-quercetin complexes???
Major remark
English require major revision. I have labeled most badly formulated thoughts.
Remarks:
1. We authors first produced apoBet v 1 “Bet v 1… was dialyzed …against …deferoxamine mesylate (DM) salt (Supplier?). Why was not Bet v 1 without DM included as controls in all experiments? Why some figures have Bet v 1 but not apoBet v 1 (Fig 2e, 3a)? Was it apoBet v1?
2. Please check all the figure legends. For example in Fig 1 … Bet v 1 binds iron-quercetin complexes (a). Protein surface …. (a) should not be before the dot but before the explanation: complexes. (a): Protein surface…
3. It is not clear what is shown in Fig. 1g. How did the authors estimate the number of iron-quercetin molecules per Bet v 1 one? (“….which allowed also (to) comput(ing) the number of bound ligands per Bet v 1 molecule….”).
4. Fig. 2c, d: what is shown in X axis? Why are the X axes different?
5. Fig. 2f: holo Bet and apoBet are not mach different, not enough to by hypoallergenic.
6. Fig. 3a: AZ-AhR cells – source?
7. Fig. 3b-e: these results are very suspicious. How long was the incubation? What was the percentage of dead cells? Either the complexes were toxic to monocytes or all that artifacts. What was the idea to study the effect on cell subpopulations?
